# Space and Vine Cultivar Interact to Determine the Arbuscular Mycorrhizal Fungal Community Composition

**DOI:** 10.3390/jof6040317

**Published:** 2020-11-27

**Authors:** Álvaro López-García, José A. Jurado-Rivera, Josefina Bota, Josep Cifre, Elena Baraza

**Affiliations:** 1Department of Animal Biology, Plant Biology and Ecology, Universidad de Jaén, E23071 Jaén, Spain; 2Department of Biology, Universitat de les Illes Balears, E07122 Palma de Mallorca, Spain; jose.jurado@uib.es (J.A.J.-R.); j.bota@uib.es (J.B.); cifre.pep@gmail.com (J.C.)

**Keywords:** *Vitis vinifera*, mycobiome, massive DNA sequencing, community assembly, spatial effects, environmental filtering

## Abstract

The interest in the use of microbes as biofertilizers is increasing in recent years as the demands for sustainable cropping systems become more pressing. Although very widely used as biofertilizers, arbuscular mycorrhizal (AM) fungal associations with specific crops have received little attention and knowledge is limited, especially in the case of vineyards. In this study, the AM fungal community associated with soil and roots of a vineyard on Mallorca Island, Spain was characterized by DNA sequencing to resolve the relative importance of grape variety on their diversity and composition. Overall, soil contained a wider AM fungal diversity than plant roots, and this was found at both taxonomic and phylogenetic levels. The major effect on community composition was associated with sample type, either root or soil material, with a significant effect for the variety of the grape. This effect interacted with the spatial distribution of the plants. Such an interaction revealed a hierarchical effect of abiotic and biotic factors in shaping the composition of AM fungal communities. Our results have direct implications for the understanding of plant-fungal assemblages and the potential functional differences across plants in vineyard cropping.

## 1. Introduction

Arbuscular mycorrhizal (AM) fungi are an important component of the plant soil microbiome. AM fungi are a low diversity monophyletic group of fungi living in close association with many terrestrial plant species [1]. The intimate association between plant and fungi occurs at the root level, and the main benefit consists of the exchange of inorganic nutrients from soil, provided by the fungus, with carbon fixed during the plant photosynthesis [2]. In addition, the symbiosis provides further advantages for the plants, allowing them to better tolerate biotic and abiotic stresses [3], thus increasing fruit yield and quality (e.g., [4,5,6]).

Due to the low diversity of AM fungi (ca. 300 described species [1]) able to colonize 80% of terrestrial plant species, the symbiosis has been considered as low specific. However, findings point towards non-random association patterns in plant-AM fungal identities [7], and the wide functional diversity exhibited by fungal taxa [8] highlights the importance of the knowledge of plant-fungal partner identity when optimizing agricultural production systems. Furthermore, it has been shown that different varieties of the same crop can respond differently to the same AM fungal isolates [9,10]. Other authors have found differences in the composition of AM fungal communities associated with different varieties, even when initially exposed to the same AM fungal community. For example, Taylor et al. [11] found that different onion genotypes displayed subtly different preferences for AM fungi from a mixed community, suggesting a selection process controlled by the plant and/or fungi (see also [12,13]). 

*Vitis vinifera* is one of the most important crops in modern agriculture. Its economic and cultural impact is undeniable [14,15], especially in the Mediterranean region which harbors 40% of the world total vineyard area [16]. The availability of water for Mediterranean agriculture is a persistent and growing problem due to climate change, representing a limitation and a threat to the future of our primary sector [17]. As shown when associated with other plants, AM fungi provide multiple functions to grapes and therefore to vineyards that can help overcome current restrictions to their cultivation: They increase grape uptake of N [18,19], reduce soil Cu toxicity [20], improve plant drought resistance [21], and protect against pests [22] (but see [23]). Moreover, it has been suggested that the whole microbiome and, particularly, AM fungi can alter the biochemical composition of grapevines [24,25], which is a key factor in the commercial market of grapevine subproducts. Despite this importance, there are few studies addressing the composition of associated AM fungal communities with vineyards. The scarce evidence suggests that AM fungal assemblages in vineyards are strongly influenced by soil type [26,27] and, at a lesser extent, by host plant [28] and management practices [29,30,31] presenting low phenological variation [28,32]. The cultivation of vineyards is done through the grafting of different cultivars onto target rootstock resistant to *Phylloxera*; therefore, several cultivars are supported by genetically identical rootstocks. However, none has focused on potential differences of grafted cultivars even when it has been demonstrated that rootstocks differ in their response to AM fungi [21]. 

Here, we focus on analyzing the AM fungal community associated with two cultivars grafted onto the same rootstock, a local cultivar name Callet and a widely-distributed one, Merlot. The use of cultivars locally adapted can be one of the major adaptation practices to face climate change as they have been described as physiologically better adapted to deficient irrigation, particularly with regard to water use efficiency [33,34]. In fact, a recent work pointed out that under similar water-deficit stress, Callet (and other local cultivars) displayed a better physiological performance compared to the widespread ones [35]. In this frame, we particularly aim to resolve the following questions:(i)Quantify the diversity of AM fungi associated with the vineyard, both those directly colonizing roots and those present at the agroecosystem level (surrounding soil);(ii)Investigate the AM fungal community composition associated with each vineyard cultivar;(iii)Quantify the relative importance of environmental/spatial factors in the distribution of AM fungal communities.

The obtained information will help to determine which characteristics of the crop may be decisive when selecting the most related AM fungal species, and also will help inform who directs the establishment of the symbiosis: Plant or fungus. If fungi drive the symbiotic relationship, the AM fungal community composition will be primarily determined by environmental/spatial factors and not by the vine cultivar. On the contrary, if the vine cultivars select their community composition, the symbiosis would be more likely driven by the plant.

## 2. Materials and Methods 

The experiment was carried out in the commercial vineyard Can Axartell in Pollença (UTM: 31S 501616.434, 4409438.756, Mallorca Island, Spain) under the appellation Vi de la Terra Mallorca, and it is organic certified. The experimental plot comprised 2.4 ha with Merlot (clone 181) and Callet (local red cultivar) cultivars, both grafted on SO4 rootstock (clone 5). The plantation was settled down in 1999 with a training system maintained as organically certified, and has a density of 3200 plants per ha (2.5 between rows and 1.25 m within rows). A drip irrigation system is available with one drip per plant (2.3 L/m^2^). The irrigation system allows adjustment according to demand, watering only in the driest months of July and August on a weekly basis.

The soil in the area comes from marls and limestone-marls of the Late Jurassic and Cretaceous periods. Quaternary sedimentary materials are also found in the area. According to Reference Soil Groups [36] the dominant soil is Calcaric Regosol. The soil is of chalky nature with a high-water retention capacity, high clay content (USDA, sand 17%, silt 31%, clay 52%), alkaline (pH 8.30), total organic carbon of 20.7 g/Kg, total nitrogen of 1.6 g/Kg, C/N ratio of 7.50, and cation exchange capacity CEC of 259.7 meq/Kg. 

The climatic conditions are typically Mediterranean, with mild winters and hot dry summers. However, as the property is located quite close to the sea (7 km), the conditions are milder than expected, both in winter and summer. Average annual rainfall is approximately 700 mm with a mean temperature of 16.5 °C.

Regarding the management of the soil in the vineyard, a spontaneous green cover is maintained in alternate rows (one not tilled, another tilled) in the alleys between vine rows (inter-rows). The cover is maintained in the central part of the inter-rows, while the vegetation between vine plants in the same row is removed several times a year by shallow cultivation in a strip about 1 m wide. In the alley, the green cover is mowed and added to the soil as a green organic fertilizer in spring. In this way, the height of the green cover manages to control the water stress in the vine plants during the end of spring and summer. The green cover is changed every several years from one row to the next, tilling the entire plot. At the time of sampling, all sampled plants were kept weed-free by shallow cultivation at the row and mowed the alleys.

### 2.1. Sampling and DNA Extraction

Soil sampling was performed 10 July 2018, during fruit development stage. Eight plants of each cultivar (either Callet or Merlot) were sampled and their spatial coordinates inside the studied plot recorded. Using a retro-scaler, a hole 40 cm in diameter was made as close as possible to the plant in the alley orientated to the north. Roots and 1 kg of soil were collected at 0–30 cm depth. Only the roots near the sampled plant were collected. The absence of surrounding weeds assured that the roots belonged to the vine. Samples were immediately placed in sterile bags and transported on ice for laboratory analyses.

Once in the laboratory, the thinnest roots were collected, discarding the thickest and oldest ones. The soil was sieved through 5.0 mm mesh, homogenized, and 20 g per sample kept at −20 °C. The roots were washed several times with abundant water until no soil was left and rinsed with distilled water. They were then cut into 0.5 cm pieces, homogenized, and stored in two 100 mg aliquots frozen at −80 °C.

Total soil DNA was extracted from samples of 0.25 g of sieved soil using the DNeasy® PowerSoil® Kit (Qiagen Inc., Mississauga, ON, Canada) in accordance with the manufacturer’s instructions. The quality of the extracts was assessed using NanoDrop spectrophotometer (Thermo Fisher Scientific, Delaware City, DE, USA) and then stored at −80 °C for further analysis.

Frozen roots were taken out of the freezer and quickly ground to a fine powder under liquid nitrogen using a mortar and pestle. Two genomic DNA extractions per root sample were performed (100 mg each) using the DNeasy Plant Mini Kit (Qiagen Inc., Mississauga, ON, Canada), following the manufacturer’s instructions.

### 2.2. DNA Amplification and Sequencing 

Extracted DNA was processed to identify AM fungal taxa by Illumina Miseq-sequencing of the 18S rRNA gene using the Glomeromycota-specific primers NS31 [37] and AML2 [38]. Library preparation and Illumina sequencing were carried out at the IPBLN Genomics Facility (CSIC, Granada, Spain). Amplicon libraries were generated through a two-step PCR strategy. The first step was carried out in a final volume of 10 µL, containing 1× KAPA HiFi HotStart ReadyMix DNA polymerase (Roche Diagnostics, West Sussex, UK), 0.2 µM forward and reverse primers, and 10 ng of the template DNA. Cycling conditions were: 95 °C 3 min, (95 °C 30 s, 58 °C 30 s, 72 °C 30 s) × 30, 72 °C 5 min. PCRs were triplicated and pooled together. A second PCR step attached dual combinatorial indices and Illumina sequencing adapters using Nextera XT v2 index kit. PCR conditions were: 95 °C 3 min, (95 °C 30 s, 55 °C 30 s, 72 °C 30 s) × 8, 72 °C 5 min. All PCRs were validated through visualization on 1.8 % (*w*/*v*) agarose gel and purified using the NucleoMag® NGS Clean-up and Size Select Kit (Macherey-Nagel, Düren, Germany). Concentrations were measured on a Qubit® fluorometer (Thermo). Amplicons were equimolarly pooled and a final library mix was run on a Bioanalyzer HS DNA chip (Agilent, Santa Clara, CA, USA) to verify quality and size distribution. The library pool was then diluted and denatured as recommended by the Illumina MiSeq library preparation guide. The 300 × 2 nt paired-end sequencing was conducted on a MiSeq sequencer. Samples were demultiplexed, and barcodes were removed and returned as individual per-sample fastq files from the sequencing facility.

### 2.3. Bioinformatic Analyses

The initial 4,617,181 MiSeq sequences were analyzed with the amplicon sequence variant (ASV, hereafter) analysis pipeline known as Divisive Amplicon Denoising Algorithm (DADA2 v. 1.8.; Ref. [39]). Briefly, forward and reverse sequences were trimmed to 295 and 290 bp, respectively. Primers were removed and a quality score set up to a minimum of 2. Sequences were dereplicated to keep unique sequences, and the error rate model inferred and used to implement the sample inference algorithm to remove Illumina sequencing errors. Forward and reverse reads were merged, and the sequence abundance table generated. Chimeric sequences based on the local dataset were removed (5.06% of quality filtered and merged reads). DADA2 gave 860 ASVs comprising 3,443,644 non-chimeric reads. The taxonomic assignment was determined for each ASV against the 16S/18S SILVA release 132 ([40] accessed 09/2018) using the RDP algorithm [41]. The SILVA database includes only a small set of representative sequences from Glomeromycota. Therefore, to improve taxonomic assignment, we amended the SILVA database with every classified sequence (i.e., those identified as a virtual taxa) in the Glomeromycotan specific database MaarjAM [42] (accessed January 2019). This database comprised 28,137 sequences including their taxonomic assignment. The taxonomic assignment was then reassessed against this combined database using RDP, and non-Glomeromycotan sequences were discarded, resulting in 636 ASVs that comprised 3,296,623 reads.

To remove further errors that were thought not to be removed by standard DADA2 pipelines, we then applied LULU algorithm [43] and obtained 240 corrected ASVs. Since DADA2-LULU infers unique original sequences in the DNA template, the result could correspond to an infra-specific level. Thus, the 240 ASVs were clustered by blasting against MaarjAM and named as the corresponding virtual taxa (VT, hereafter) when showing an identity higher than 97%. Those ASVs with low query cover (<90%) or low E-value during the blast were discarded (4/240). ASVs non-fitting at a minimum of 97% were aligned together with the rest of ASVs using MAFFT [44] and clustered at 97% using VSEARCH [45] implemented in MOTHUR [46]. Those ASVs clustering with VT-named ASVs were added to the existing cluster and those clustering alone were considered as new VTs. With 3,295,222 reads, we finally obtained 56 VT, five of which were defined as new VT non-included in MaarjAM database. The bioinformatic pipeline is available as Appendix A.

Data were deposited and are available in the Sequence Read Archive under Bioproject ID PRJNA679172. Representative sequences of the detected VT were deposited in GenBank under the accession numbers MW285643-MW285698. Processed raw data is available as Appendix A. 

### 2.4. Statistical Analyses

Prior to subsequent analyses, the sequencing information of the two subreplicates per root sample were pooled into a unique sample. The diversity coverage of the sequencing was checked by visualizing rarefaction curves by means of the *rarecurve* function (*vegan* R package [47]).

To build the VT abundance matrix, read counts per VT and sample were used as a proxy of abundance. The VT abundance matrix was subjected to Hellinger transformation for subsequent analyses [48]. 

The VT abundance matrix was relativized to total row sums and used to obtain VT richness, Simpson (1-D) dominance, and Shannon indices at a sample level. The phylogenetic diversity was obtained by calculating the standardized effect size of the mean pairwise phylogenetic distance (ses.mpd) of AM fungal communities [49]. For that, the most abundant ASV per VT was selected as representative sequences and aligned using MAFFT 7.0. The Tamura-Nei nucleotide substitution model with a discrete gamma distribution was found to be the best fitted using MEGA X [50], and it was used to correct the evolutionary distance matrix between aligned sequences. The ses.mpd was calculated using the VT abundance matrix plus the evolutionary distance matrix of the VTs in each sample and compared to 999 null communities obtained using the independent swap algorithm which maintains species occurrence frequency and sample species richness (*ses.mpd* function, *picante* R package [51]). The mean values of ses.mpd per treatment were then used to judge the clustering or segregation against null communities. Significance of the calculated index was assessed with a *t*-test.

The spatial distribution of samples was decomposed via principal coordinates of neighbor matrices (PCNM). The significance of PCNM axes with positive eigenvalues on VT abundance distribution was evaluated by means of permutational multivariate analysis of variance (PERMANOVA, McArdle and Anderson 2001, *adonis* function, *vegan* R package [47]), using 999 permutations and Euclidean distance as measures of dissimilarity. This dissimilarity, as the abundance matrix was Hellinger-transformed, must be considered a Hellinger-based dissimilarity [48]. Those non-significant axes were discarded from further analyses.

The impact of vineyard cultivar, sample type (either soil or root), and their interaction on the mentioned diversity indices was tested via generalized least-squares models (*gls* function, *nlme* R package [52]), with variance structures applied for the sample type using the *varIdent* function to account for heteroscedastic variance in the model residual [53]. The models were also tested including spatial autocorrelation as covariate, both the *x* and *y* coordinates, and the selected first axis of PCNM decomposition.

The effects of the experimental variables: Cultivar, sample type, and spatial position (PCNM axes), and their interaction on community composition was addressed by means of PERMANOVA (999 permutations, Euclidean distance). As PERMANOVA is sensitive to changes in multivariate dispersion among samples, the *betadisper* function (*vegan* R package) was used to assess differences in multivariate dispersion across factor levels. When finding significant interactions across explanatory variables, partial PERMANOVAs were run for each factor level. In the case of PCNM axes, their values were split in quartiles and analyzed in four different PERMANOVAs. To visualize the found patterns, we used a redundancy analysis (RDA) [54] ordination plot constraining by the mentioned experimental variables (*rda* function, *vegan* R package).

AM fungal VT indicative of particular groups of samples were identified using Dufrêne-Legendre indicator species analysis [55], implemented by the *indval*() function (*labdsv* R package [56]).

The R script containing the applied statistical analysis is available as Appendix A.

## 3. Results

### 3.1. Sequencing Data

From the initial 4,617,181 raw reads, 3,295,222 reads passed the quality filters and were found to belong to Glomeromycota. They were assigned to 56 VT, including five novel VTs not included in the MaarjAM database (see the phylogenetic tree, Appendix A): Two Paraglomeraceae and three Glomeraceae. The 56 virtual taxa belonged to eight Glomeromycotan families, Glomeraceae being the most abundant (88.94% of the reads), followed by Claroideoglomeraceae (6.86%), Paraglomeraceae (2.83%), Diversisporaceae (1.01%), Archaeosporaceae (0.33%), Acaulosporaceae (0.02%), and Gigasporaceae and Ambisporaceae (both of them with <0.01%) (Table A1). Glomeraceae was dominant both in soil and root samples; however, this family reached up to 97.43% of reads in the latter while the abundance across families was more spread in soil samples (72.89% for Glomeraceae in that case). For all samples, the sampling effort curve showed a saturation of VTs with an increasing number of sequences (see rarefaction curves, Appendix B
Figure A1), ensuring that the sequencing effort captured the AM fungal diversity of the samples.

### 3.2. AM Fungal Diversity

Only sample type between the explanatory variables was found to affect the taxonomic and phylogenetic diversity indices (Table 1), in general showing an increase in the samples with a soil-borne origin (Figure 1). Implementation of spatial autocorrelation in the models did not improve the explanatory power of the diversity indices (analyses not shown).

On average, we found 27.25 (±0.85 S.E.) VT per sample (29.31 ± 0.96 for soil and 25.18 ± 0.53 for root), ranging between a minimum of 18 and a maximum of 36. In the case of Shannon and Simpson (1-D) indices, soil and root samples showed similar patterns being higher and less variable in the case of soil (Shannon 2.58 ± 0.03; Simpson 0.89 ± 0.00) than in roots (2.00 ± 0.05 and 0.78 ± 0.02, respectively). Regarding the ses.mpd index, root samples had on average smaller and more negative values than soil samples (−1.21 ± 0.11 versus 0.63 ± 0.17) and they were found to be significantly different from the generated null values (*t* = −7.694, *p* < 0.001). This means that the AM fungal communities associated with roots exhibited a significant phylogenetic clustering. Conversely, the positive values shown by soil samples also differed from null expectations (*t* = 2.654, *p* = 0.018), indicating phylogenetic overdispersion.

### 3.3. Drivers of AM Fungal Community Composition

Between the generated PCNM axes, only PCNM1 was found to impact AM fungal community composition (*F* = 1.996, *R*^2^ = 0.061, *p* = 0.048; Table A2) and hence was fed into the subsequent analyses. This means that the AM fungal communities tended to vary at the roughest calculated spatial scale in the studied area as far as the PCNM decomposition advanced from the widest to the smallest scale (see Figure 2; Ref. [57]).

The PERMANOVA model showed a significant effect on AM fungal community composition of the sample type (*F* = 11.971, R^2^ = 0.269, *p* < 0.001; Table 2). No difference in multivariate dispersion was found across sample types. This pattern was easily visualized in the RDA ordination plot (RDA model *F* = 5.403, *p* = 0.001; Figure 3) where the first axis (explaining ca. 30% of AM fungal community variance) clearly separated root from soil samples. In agreement, a series of AM fungal VT were recorded as indicators of soil and root samples (16 and 5 out of 56 recorded VT, respectively, Table A3). It is noteworthy that the five VT tied to roots belonged to Glomeraceae family; meanwhile, the 16 tied to soil were spread across Archaeosporaceae, Claroideoglomeraceae, Diversisporaceae, Glomeraceae, and Paraglomeraceae.

Space showed an impact on AM fungal community composition alone and by interacting with the cultivar (*F* = 2.787, *R*^2^ = 0.063, *p* = 0.011 and *F* = 2.086, *R*^2^ = 0.047, *p* = 0.034, respectively). The effect of space can also be seen in the RDA ordination as far as PCNM1 aligns well with the second axis of the ordination (5.6% explained variance). The cultivar was not significant, apart from its interaction with space. Nevertheless, the species indicator analyses detected four VT tied to either Callet or Merlot cultivars (Table A3). When PERMANOVA was run separately by cultivar, space (PCNM1) was found to significantly influence AM fungal community composition in both cases, Caller and Merlot (Table 3a). Alternatively, when the analysis was run on the highest and lowest values of PCNM1 separately, the cultivar only drove the AM fungal community in the case of values over the median (Table 3b): i.e., yellowish positions in Figure 2. No differences in multivariate dispersion were detected in these analyses.

## 4. Discussion

### 4.1. Diversity of Vineyard Associated AMF

The VT richness found in the present study was similar to that found in other previous studies of AM fungi using massive sequencing approaches, either in Mediterranean natural ecosystems [58,59] or in vineyards [27,29]. However, comparisons across studies of vineyards are difficult due to the scarcity and diversity of molecular analyses on AM fungi developed until now. As far as we know, only one publication uses Illumina MiSeq technology, incorporating the most widely used ribosomal region (18S) (see [42,60] for the analysis of AM fungi). The distribution of AM fungal families found in our results resembles the one found by Vukicevich et al. [31] and that found using 454-pyrosequencing in other ribosomal regions [61]. This distribution is based on the dominance of the Glomeraceae family, followed by Claroideoglomeraceae, and the rest of the families in a more marginal abundance. It is noteworthy that the wide diversity of AM fungi found in our study (up to eight families) was high in the vineyard system, in contrast with previous studies, e.g., [61]. The relationship between phylogeny and symbiotic functioning of AM fungal groups has been widely observed and linked to the functional traits they exhibit [8,62,63]. In this regard, functional differences have been mainly investigated across four main families: Glomeraceae, Gigasporaceae, Acaulosporaceae, and Claroideoglomeraceae. Glomeraceae members are identified as wide root colonizers with improved ability to confer resistance against pathogens and drought, but smaller P uptake for the plant (in comparison with Gigasporaceae) [63,64]. However, the functionality within Glomeraceae still deserves to be studied due to the wide number of species it harbors and the found contradictory results (e.g., Yang et al. [64] found increased P uptake for Glomeraceae). Gigasporaceae is characterized by producing extensive hyphae into the soil, benefitting soil aggregation and P uptake, but showing a smaller capacity of hyphal healing [65]. The latter characteristic probably makes them more likely to disappear when environmental conditions are not optimal [66]. Acaulosporaceae has been linked to stress-tolerant strategies showing poor colonization ability, both in soil and roots [67], and Claroideoglomeraceae has shown increased capacity to provide resistance against nematodes [64]. 

Despite the scarcity of knowledge on functionality of other glomeromycotan families, the high phylogenetic diversity found in our study indicates that this vineyard agrosystem presents a considerable functional diversity that can be translated into the provision of multiple ecosystem services by AM fungi. Indeed, we found representatives of eight out of ten accepted glomeromycotan families in the MaarjAM database belonging to the four described orders [42]. It is generally accepted that agrosystems usually harbor a decreased AM fungal diversity, often lacking important members of the AM fungal phylogeny and being composed primarily of Glomeraceae (see [66,68]). In our case, it seems the system harbors a representation of the full phylogenetic diversity of AM fungi. This wider phylogenetic diversity associated with plants with long lifespans and higher competitor abilities, as a vine is in comparison with annual crops, has been proposed and demonstrated before [59]. In agreement, another study found no differences in the diversity levels of AM fungal communities in vineyards when comparing with adjacent natural areas [69].

We detected five AM fungal virtual taxa out of 56 non-previously recorded in the MaarjAM database. This value does not differ from those found in other studies in South Spain: [59] found four novel VT out of 84, and [58] found 31 out of 96. It is possible that the insularity of our study site affected the diversity, increasing the degree of endemism. However, we should have recorded more new VT than has been found in other Mediterranean natural areas. Our results confirm the found lack of island biogeography in the diversity of the AM fungal groups on Mallorca [70].

### 4.2. Effect of Sample Type

We found a marked difference between the recorded AM fungal community composition and diversity in roots of vines and the surrounding soil. This was expected due to the previous knowledge about life history strategies of AM fungi, which states that members of the Glomeraceae family primarily colonize the inner root in comparison with other AM fungal families (e.g., Acaulosporaceae and Gigasporaceae [62]). This agrees with the fact that AM fungal phylogenetic diversity in the same system is usually higher in soil than in roots [58,71] and this has also been found in vineyard studies when comparing vine roots and the spore community [28]. 

Nevertheless, we cannot ignore that our increased phylogenetic diversity in the soil is the result of the presence of accompanying spontaneous vegetation. Although weeds were not present at the time of sampling, soil management on the farm under study allows the growth of spontaneous vegetation during part of the year. Due to the partner preferences in the AM symbiosis, higher plant diversities usually support higher diverse AM fungal communities (e.g., [7,72]). In the particular case of cover crops, their role has been highlighted as a way to recruit higher soil microbial diversity to increase the stability and functional properties of the system [73]. Indeed, some studies in vineyards have pointed out that herbaceous weed species are supporters and providers of higher AM fungal diversities [27,74,75]. In fact, the diversity of AMF determined by pyrosequencing was greater in a covered vineyard than in a tilled vineyard [29]. Our data show that it is not necessary a continuous maintenance of the cover. The maintenance of green cover during periods of time that limit its competition with the vineyard would also allow high levels of AMF diversity in the soil of the agroecosystem to be maintained. Moreover, the organic management of the vineyard may have contributed to the increased phylogenetic diversity of AM fungi.

### 4.3. Interactive Effect of Cultivar and Space

The most interesting result of the current study was the finding that the vine cultivar, even when grafted onto the same rootstock, drove the community composition of AM fungi. Previous studies that have found effects of the plant species genotype on the associated soil microbial communities (e.g., [76,77,78]) attribute differences to the rhizodeposits released by the plants, even at the genotype level of the same species [79]. Regarding vine, some previous studies have stated differences in soil microbial communities associated with different rootstock genotypes [80,81]. In the particular case of AM fungi, no effect of vine cultivar has been found either at rootstock [61] or cultivar [27]. We did find differences between cultivars in agreement with other studies reporting differences in AM fungal communities associated with other plant species genotypes (e.g., [11,82]). In our case, the found explained variation was relatively low (ca. 5%). However, when comparing with other studies looking for differences in AM fungal community composition across plant species, the magnitude seems similar: e.g., Varela–Cervero et al. [58] reported 8% of variation attributed to plant species, and Sepp et al. [7] 16%. Nevertheless, it has also been argued that the effect of host genotype on the composition of rhizosphere microbial communities is usually smaller in agricultural systems than in ecosystems with long-term coevolution of plant-microbial interactions, i.e., natural systems [79].

We found a significant portion of variance explained by the spatial position of the samples. In the context of ecological community assembly, spatial autocorrelation can be interpreted as an effect of dispersal events [83]. However, these patterns can be confounded with unmeasured environmental variables [84]. Although we did not record soil variables in the current study, the existence of a background soil environmental gradient seems very likely. AM fungal communities are usually shaped by soil variables such as pH, organic matter content or soil phosphorus [30,85]. Hence, a spatially autocorrelated soil environmental gradient could explain why vine cultivar interacted with space to shape the AM fungal community. In this sense, the soil gradient could imply a strong environmental filtering in one extreme, limiting the diversity of AM fungi and impeding vine cultivars to select for different AM fungal communities. Alternatively, we could record differences between cultivars in the other extreme of the gradient as we found. The hierarchy of assembly mechanisms of biological communities is a well-stated fact that locate environmental filtering at a broader spatial scales and biological interactions (as partner selection in symbiosis) at a finer level [86,87], as we found.

## 5. Conclusions

As agricultural practices advance towards more sustainable production, study and interest in the role and functions of plant-associated microbiomes have increased [88,89]. Soil is a pivotal component of the ecosystem and generally acts as a microbial reservoir for plants [90,91]. We have revealed interesting patterns in the AM fungal communities associated with an economically valuable crop. On one hand, we recorded a high microbial diversity that should be considered when evaluating ecosystem services associated with this crop. Moreover, the reasons behind this trend should be studied in depth to better assess the most beneficial crop management, for example in the case of cover vegetation. On the other hand, we revealed differences in the AM fungi associated with the different vine cultivars. Given that even small host genotype-mediated effects on microbiome composition can have large effects on host health [92], this pattern needs to be further evaluated, perhaps to dig into potential functional differences on the microbiota-extended plant phenotypes.

## Figures and Tables

**Figure 1 jof-06-00317-f001:**
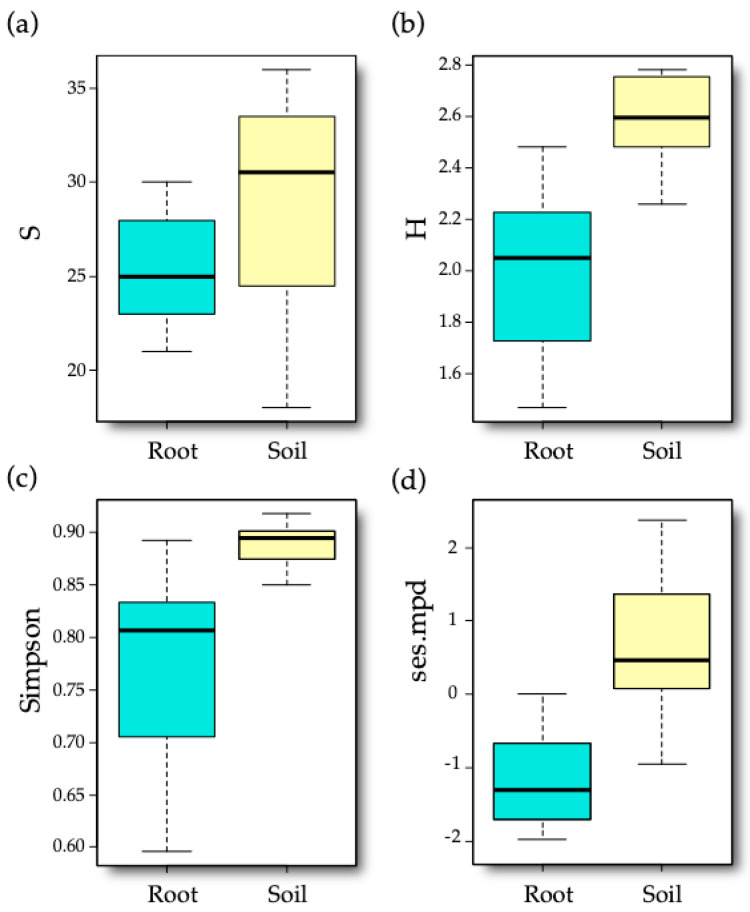
Boxplots showing the distribution of diversity indices by sample type. (**a**) VT richness (S). (**b**) Shannon diversity index (H). (**c**) Simpson (1-D) diversity index. (**d**) Standardized effect size of mean pairwise phylogenetic distance (ses.mpd).

**Figure 2 jof-06-00317-f002:**
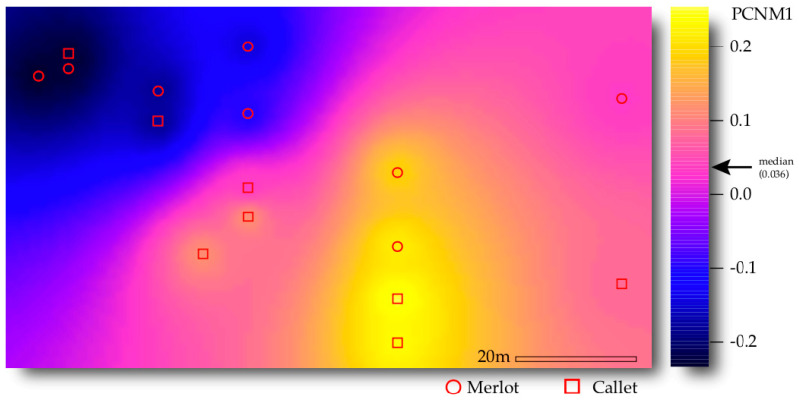
Spatial distribution of the first axis of the principal coordinates of neighbor matrices’ (PCNM) spatial decomposition over the studied plot. The values were interpolated across sampled points. The colored scale indicates the PCNM values.

**Figure 3 jof-06-00317-f003:**
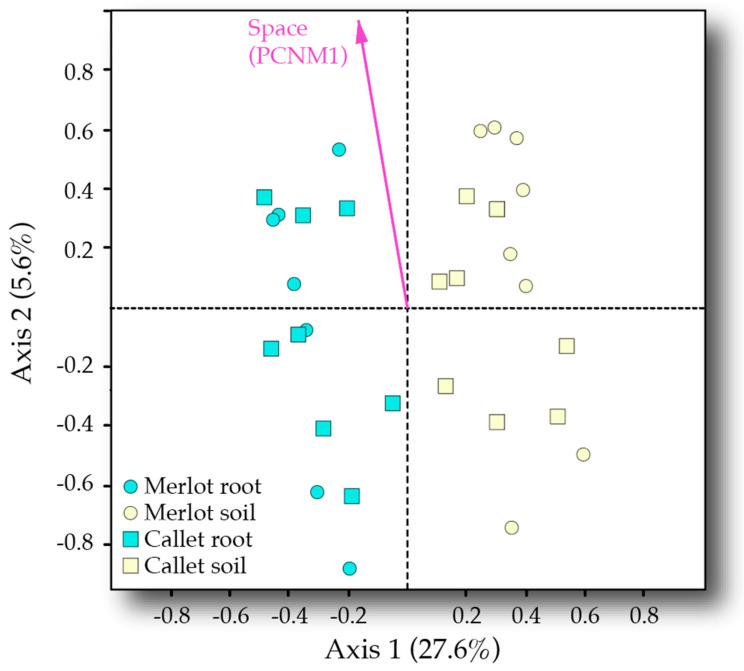
RDA ordination of AM fungal communities constrained by space (PCNM first axis), sample type (root or soil), and cultivar.

**Table 1 jof-06-00317-t001:** Linear models showing the effect of sample type (soil vs. root), cultivar, and their interaction on diversity indices of the arbuscular mycorrhizal (AM) fungal communities. *F* values and degrees of freedom (as subscripts) are shown. Bold letters indicate significant variables: * *p* < 0.05; *** *p* < 0.001.

Diversity Index	Sample Type	Cultivar	Sample Type × Cultivar
VT richness	**6.803_1,28_ ***	0.619_1,28_	0.756_1,28_
Shannon	**46.348_1,28_ *****	2.719_1,28_	0.364_1,28_
Simpson	**27.240_1,28_ *****	1.98_1,28_	1.43_1,28_
Ses.mpd	**43.968_1,28_ *****	0.721_1,28_	2.173_1,28_

**Table 2 jof-06-00317-t002:** Effect of sample type (soil vs. root), cultivar, spatial position (PCNM first axis), and their interactions with AM fungal community composition (permutational multivariate analysis of variance (PERMANOVA) 999 permutations and Euclidean distance as measure of dissimilarity). Bold values indicate significance.

Variable	*Df*	*SS*	*MS*	*F*	*R* ^2^	*p*
Sample type	1	3.401	3.401	11.971	0.269	**<0.001**
Cultivar	1	0.444	0.444	1.564	0.035	0.117
Space (PCNM1)	1	0.792	0.792	2.787	0.063	**0.011**
Sample type × Cultivar	1	0.222	0.222	0.782	0.018	0.617
Sample type × Space (PCNM1)	1	0.234	0.234	0.824	0.019	0.559
Cultivar × Space (PCNM1)	1	0.593	0.593	2.086	0.047	**0.034**
Sample type × Cultivar × Space	1	0.143	0.143	0.502	0.011	0.942
Residuals	24	6.819	0.284		0.539	
Total	31	12.648			1.000	

**Table 3 jof-06-00317-t003:** Separate PERMANOVAs for cultivar (**a**) or PCNM1 values above and below the median value (**b**); 999 permutations and Euclidean distance as measure of dissimilarity. Bold values indicate significance.

(**a**)							
**Data subset**							
	**Variable**	***Df***	***SS***	***MS***	***F***	***R*^2^**	***p***
Callet	Space (PCNM1)	1	0.598	0.598	1.947	0.101	**0.043**
	Sample type	1	1.421	1.421	4.627	0.239	**0.001**
	PCNM × Sample type	1	0.236	0.236	0.768	0.040	0.677
	Residuals	12	3.686	0.307		0.620	
	Total	15	5.942			1.000	
Merlot	Space (PCNM1)	1	0.786	0.786	3.012	0.126	**0.016**
	Sample type	1	2.202	2.202	8.436	0.352	**0.001**
	PCNM × Sample type	1	0.141	0.141	0.539	0.022	0.851
	Residuals	12	3.133	0.261		0.500	
	Total	15	6.262			1.000	
(**b**)							
PCNM1 >0.036	Cultivar	1	0.590	0.590	2.210	0.102	**0.031**
	Sample type	1	1.808	1.808	6.774	0.314	**0.001**
	Sample type × Cultivar	1	0.158	0.158	0.592	0.027	0.816
	Residuals	12	3.203	0.267		0.556	
	Total	15	5.759			1.000	
PCNM1 <0.036	Cultivar	1	0.308	0.308	0.947	0.050	0.441
	Sample type	1	1.758	1.758	5.402	0.287	**0.002**
	Sample type × Cultivar	1	0.154	0.154	0.472	0.025	0.948
	Residuals	12	3.905	0.325		0.638	
	Total	15	6.124			1

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
