# Peer review of "Space and Vine Cultivar Interact to Determine the Arbuscular Mycorrhizal Fungal Community Composition"

_jof, 2020, doi:10.3390/jof6040317_

Round 1

Reviewer 1 Report

The authors clearly presented their work. Quantification of AM diversity in soil and plant, AM community composition in relation to vine cultivar and AM community distribution in relation to spatial factors have been put into focus.  The sequencing data revealed rich microbial fungal diversity, the prevalence of Glomeraceae in soil and root, with soil containing a wider AM diversity then roots. The paper generates interesting conclusion but also raises many questions about the patterns on the AM communities association to grapevine plants. The most interesting outcome is that space and vine cultivar even when grafted on the same rootstock determine the AM composition. Methodology and findings described here are beneficial to scientific public so my recommendation is to accept the paper in present form.

Author Response

We are very thankful for the positive comments and we are glad that the reviewer see merit in our research.

Reviewer 2 Report

Review: jof-968453

The MS “Space and vine cultivar interact to determine the arbuscular mycorrhizal fungal community composition” deals with the effects of different cultivars on the fungal communities.

The topic is consistent with the aims and scopes of the journal and the paper is interesting. However, it has some errors that need to be corrected or explained in order to make the results better understood.

I recommend it for publication consideration after a careful minor revision. Please find below some comments:

Line 57: Despite or in spite of, please, don’t combine

Lines 65-67: I suggest deleting this information. I don’t think it is accurate.

Lines 67-68: Please add some scientific support for this info.

Line 69: replace “joint to” with “and”.

Line 70: delete “as”

Lines 70-71: consider deleting

Line 78-83: Revise grammar. It is hard to understand.

Line 87: space after 2.4 ha

Line 88: “settle down”, settle means different.

Line 106, etc… unify the “°C” terminology.

Lines 103-104: Revise grammar. Hard to understand

Line 116: replace “with the help” with “using”

Lines 185-186: I strongly advise to add information. Given the importance of different AMF communities presence due to it is known that may affect the symbiosis, I think it would be interesting to discuss (and maybe deepen into) the contribution of each family found in the vineyard and their potential effects.   

Author Response

The MS “Space and vine cultivar interact to determine the arbuscular mycorrhizal fungal community composition” deals with the effects of different cultivars on the fungal communities.

The topic is consistent with the aims and scopes of the journal and the paper is interesting. However, it has some errors that need to be corrected or explained in order to make the results better understood.

I recommend it for publication consideration after a careful minor revision. Please find below some comments:

Response: we thank for the positive comments. Despite the specific comments made by the reviewer, an English native speaker (dr. Jennifer A. Krumins) has revised the text and suggested grammatical changes to improve the quality of language.

Line 57: Despite or in spite of, please, don’t combine

Response: changed to "Despite this importance,..."

Lines 65-67: I suggest deleting this information. I don’t think it is accurate.

Lines 67-68: Please add some scientific support for this info.

Line 69: replace “joint to” with “and”.

Line 70: delete “as”

Lines 70-71: consider deleting

Response: The entire paragraph (now in lines 64-70) has been updated to include the five comments. Now it reads:

"Here, we focus in analyzing the AM fungal community associated to two cultivars grafted onto the same rootstock, a local cultivar name Callet and a widely-distributed one, Merlot. The use of cultivars locally adapted can be one of the major adaptation practices to face climate change as they have been described as physiologically better adapted to deficient irrigation, particularly with regard to water use efficiency [33, 34]. In fact, a recent work pointed out that under similar water deficit stress, Callet (and other local cultivars) displayed a better physiological performance compared to the widespread ones [35]. In this frame, we particularly aim to resolve the following questions:"

Line 78-83: Revise grammar. It is hard to understand.

Response: the paragraph has been changed to: 

"The obtained information will help to determine which characteristics of the crop may be decisive when selecting the most related AM fungal species, and also will help inform who directs the establishment of the symbiosis: plant or fungus. If fungi drive the symbiotic relationship, the AM fungal community composition will be primarily determined by environmental/spatial factors and not by the vine cultivar. On the contrary, if the vine cultivars select their community composition, the symbiosis would be more likely driven by the plant."

Line 87: space after 2.4 ha

Response: changed accordingly. We have also changed this issue all along the manuscript.

Line 88: “settle down”, settle means different.

Response: changed accordingly.

Line 106, etc… unify the “°C” terminology.

Response: changed accordingly.

Lines 103-104: Revise grammar. Hard to understand

Response: changed to:

"The cover is maintained in the central part of the interrows, while the vegetation between vine plants in the same row is removed several times a year by shallow cultivation in a strip about 1 m wide."

Line 116: replace “with the help” with “using”

Response: changed accordingly.

Lines 185-186: I strongly advise to add information. Given the importance of different AMF communities presence due to it is known that may affect the symbiosis, I think it would be interesting to discuss (and maybe deepen into) the contribution of each family found in the vineyard and their potential effects.

Response: We thank for the suggestion. We have complemented the diversity section in Discussion (L330-344) to make specific references to the knowledge of functions of specific families of AMF. Now it reads as follows:

"It is noteworthy that the wide diversity of AM fungi found in our study (up to eight families) is high in the vineyard system, in contrast with previous studies [e.g. 61]. The relationship between phylogeny and symbiotic functioning of AM fungal groups has been widely observed and linked to the functional traits they exhibit [8, 62, 63]. In this regard, functional differences have been mainly investigated across four main families: Glomeraceae, Gigasporaceae, Acaulosporaceae and Claroideoglomeraceae. Glomeraceae members are identified as wide root colonizers with improved ability to confer resistance against pathogens and drought, but smaller P uptake for the plant (in comparison with Gigasporaceae) [64]. However, the functionality within Glomeraceae still deserves to be studied due to the wide number of species it harbors and the found contradictory results (e.g. Yang et al. [64] found increased P uptake for Glomeraceae). Gigasporaceae is characterized by producing extensive hyphae into the soil, benefitting soil aggregation and P uptake, but showing a smaller capacity of hyphal healing [65]. The latter characteristic probably makes them more likely to disappear when environmental conditions are not optimal [66]. Acaulosporaceae has been linked to stress-tolerant strategies showing poor colonization ability, both in soil and roots [67], and Claroideoglomeraceae has shown increased capacity to provide resistance against nematodes [64]."